# Synergistically Anti-Multiple Myeloma Effects: Flavonoid, Non-Flavonoid Polyphenols, and Bortezomib

**DOI:** 10.3390/biom12111647

**Published:** 2022-11-07

**Authors:** Kaixi Ding, Wei Jiang, Huanan Jia, Ming Lei

**Affiliations:** Hospital of Chengdu University of Traditional Chinese Medicine, Chengdu 610075, China

**Keywords:** multiple myeloma, bortezomib, flavonoids, non-flavonoid polyphenols, synergy

## Abstract

Multiple myeloma (MM) is a clonal plasma cell tumor originating from a post-mitotic lymphoid B-cell lineage. Bortezomib(BTZ), a first-generation protease inhibitor, has increased overall survival, progression-free survival, and remission rates in patients with MM since its clinical approval in 2003. However, the use of BTZ is challenged by the malignant features of MM and drug resistance. Polyphenols, classified into flavonoid and non-flavonoid polyphenols, have potential health-promoting activities, including anti-cancer. Previous preclinical studies have demonstrated the anti-MM potential of some dietary polyphenols. Therefore, these dietary polyphenols have the potential to be alternative therapies in anti-MM treatment regimens. This systematic review examines the synergistic effects of flavonoids and non-flavonoid polyphenols on the anti-MM impacts of BTZ. Preclinical studies on flavonoids and non-flavonoid polyphenols-BTZ synergism in MM were collected from PubMed, Web of Science, and Embase published between 2008 and 2020. 19 valid preclinical studies (Published from 2008 to 2020) were included in this systematic review. These studies demonstrated that eight flavonoids (icariin, icariside II, (-)-epigallocatechin-3-gallate, scutellarein, wogonin, morin, formononetin, daidzin), one plant extract rich in flavonoids (Punica granatum juice) and four non-flavonoid polyphenols (silibinin, resveratrol, curcumin, caffeic acid) synergistically enhanced the anti-MM effect of BTZ. These synergistic effects are mediated through the regulation of cellular signaling pathways associated with proliferation, apoptosis, and drug resistance. Given the above, flavonoids and non-flavonoid polyphenols can benefit MM patients by overcoming the challenges faced in BTZ treatment. Despite the positive nature of this preclinical evidence, some additional investigations are still needed before proceeding with clinical studies. For this purpose, we conclude by providing some suggestions for future research directions.

## 1. Introduction

Multiple myeloma(MM) is a clonal plasma cell tumor originating from the post-germinal lymphoid B cell line. It has an incidence of approximately 2.1 out of 100,000 people worldwide [1]. Its incidence varies notably among different races, with older individuals at higher risk [2]. Moreover, it often leads to poor outcomes in low and middle-income countries due to the lack of specialized health care, diagnosis, and advanced treatments. The 5-year survival rate for MM patients treated in Nigeria is only 7.6% [3].

At present, internal medicine intervention predominates in MM therapy. Standard treatment options include (1) induction therapy with a combination of injectable proteasome inhibitors(i.e., bortezomib(BTZ)), oral immunomodulators(i.e., lenalidomide), and dexamethasone; and (2) autologous stem cell transplantation(ASCT) with subsequent maintenance lenalidomide [4]. According to the clinical evidence, Velcade, Revlimid, and Dexamethasone (VRD)-based pre-ASCT induction therapy enhances the prognosis of MM patients [5,6]. Nevertheless, these interventions remain challenged by MM’s malignant qualities. For example, ASCT is not recommended for MM patients who are over the age of 65 or who have severe comorbid conditions like renal or pulmonary impairment, cardiac disease, or hepatic disease [7]. Most significantly, intrinsic clonal heterogeneity and genomic instability of plasma cells influence both inherent and acquired drug resistance [8]. For most MM patients, after several remissions and relapses, drug resistance develops into almost all available therapies [9]. Additionally, these MM medications could also have adverse effects. BTZ, for instance, may have adverse effects on the cardiovascular system, bone marrow suppression, and peripheral neuropathy [10]. Therefore, in order to overcome these difficulties, alternative and supportive therapies are required.

In recent years, natural dietary compounds have attracted the attention of researchers. They have a wide range of biological activities, such as anti-inflammatory, antioxidant, anti-aging, immune enhancement, as well as anti-tumor [11,12,13,14,15]. Among them, flavonoids, a class of polyphenolic plant secondary metabolites, are of great interest. Through inhibition of cell proliferation, stimulation of apoptosis, targeting of cancer stem cells, inhibition of angiogenesis and metastasis, and chemotherapy sensitization, flavonoids have anti-cancer effects [16]. In addition, the cancer prevention and treatment potential of some non-flavonoid polyphenols are equally noteworthy [17,18]. Jöhrer and Pojero reviewed the tumor activity inhibition and targeting cancer effects of some natural compounds (including polyphenols) in MM in preclinical trials, respectively [19,20]. As mentioned earlier, the main challenge MM drug interventions face is drug resistance. Therefore, exploring possible alternative treatments to overcome it is clinically valuable. This systematic review focuses on the synergistic anti-MM effects of flavonoids, other non-flavonoid polyphenols, and BTZ.

## 2. Research Methodology

The systematic review adheres to the guidance of the Preferred Reporting Items for Reference Systems Evaluation and Meta-Analysis (PRISMA) statement [21]. Preclinical studies on flavonoids or non-flavonoid polyphenols -bortezomib synergistic potentiation of anti-MM effects were collected by searching PubMed, Web of Science(WOS), and Embase for the keywords “flavonoid” or “polyphenol,” “bortezomib” or “Velcade,” and “Multiple myeloma” or “Myeloma, Plasma-Cell.”

Searching with the search formula “Multiple myeloma” or “Myeloma, plasma cell” (subject) and “Bortezomib or Velcade” (all fields) and “flavonoids or polyphenols” (all fields) were put into WOS, 33 publications were found. Performing a search at PubMed with the search formula: ((Multiple myeloma or “Myeloma, Plasma-Cell”) AND (Bortezomib or Velcade)) AND (flavonoids or polyphenols), 38 publications were retrieved. Searching in Embase with the search terms: ((Multiple myeloma or “Myeloma, Plasma-Cell”) AND (Bortezomib or Velcade)) AND (flavonoids or polyphenols), 144 publications were retrieved. The filters had not been applied in any of the above three databases. Endnote 20 was used to check for duplicates automatically and by hand for 215 publications, and 41 duplicates were taken out. The titles and abstracts of 174 publications were reviewed by two independent reviewers (Kaixi Ding and Wei Jiang). In case no consensus was reached, a third independent reviewer made the final decision. 121 publications that were not relevant to the topic of this review were removed.

The remaining 53 publications were then read in full by two independent reviewers. According to the inclusion and exclusion criteria: (1) treatment with flavonoid compounds or non-flavonoid polyphenolic compounds or plant extracts containing polyphenols; (2) animal studies or cell cultures; (3) interventions in animals or cell models of multiple myeloma; (4) studies included flavonoid compounds or non-flavonoid polyphenolic compounds or plant extracts containing polyphenols and BTZ synergistic anti-MM effects or mechanisms (5) Excluding review literature, meta-analyses, case reports, editorials, abstracts and conference proceedings and other types of literature. A total of 38 publications were discharged after a detailed review by the reviewers. In addition, two reviewers agreed that four publications with relevant citation searches from the research and the review literature should be added as secondary sources.

Eventually, 19 valid articles (Published from 2008 to 2020) were included in this review (Figure 1) (Table 1).

## 3. Mechanisms of MM

MM’s onset, progression, metastasis, and osteolytic destruction are associated with the regulation of multiple signaling pathways (Figure 2). Throughout the development of MM, rapid cell proliferation was mainly regulated by PI3K/Akt, Ras/Raf/MEK/Erk, JAK/STAT, Wnt/β-catenin, and RANK/RANKL/OPG signal pathways [41]. Uncontrolled cell proliferation was attributed to the inhibition of normal cell cycle regulation (e.g., FOXO1 and Cyclin D1), and the upregulation of downstream key anti-apoptotic factors (e.g., Bcl2, Bcl-xl, Mcl-1, and Caspase) [42]. Angiogenesis, mainly promoted by VEGF, provides oxygen and nutrients for tumor proliferation. Additionally, the activation of the Wnt/-catenin and RANK/RANKL/OPG signal pathways is primarily responsible for osteolytic bone disease in MM, which is brought on by an increase in osteoclast activity [41].

Finally, MM cells can further develop drug resistance, and this may be related to mechanisms such as the upregulation of permeability-glycoprotein(P-gp) (leading to reduced drug aggregation in vivo). IL-6 and insulin-like growth factor (IGF), two cytokines produced by bone marrow mesenchymal stem cells, also encourage drug resistance in MM cells(activating specific signaling pathways that lead to drug resistance) [43].

## 4. Bortezomib in MM Therapy

An important class of medications for the treatment of MM is protease inhibitors. Based on targeted inhibition of the 26S proteasome, protease inhibitors are crucial in the pathogenesis and proliferation of MM [44]. BTZ, the first protease inhibitor, approved for clinical use in 2003, has resulted in gains in overall survival, progression-free survival, and remission rates in patients with MM (Figure 3) [45]. Its main anti-cancer mechanism is the inhibition of the chymotrypsin-like site of the 20S protein hydrolysis core within the 26S proteasome, which induces cell cycle arrest and apoptosis [46]. The main signaling mechanisms of BTZ-induced apoptosis in MM cells are NF-κB blockade and JNK activation. In addition, BTZ stabilizes various tumor suppressor proteins, such as P53, inhibiting MM cell cycle progression [47]. Currently, the drug is commonly used in MM patients in first-line, relapsed, and/or refractory settings [48]. However, drug resistance in MM therapy is the main challenge currently faced when using BTZ [43].

## 5. Exploration of Synergistic Drug Combinations in MM Therapy

When two or more medications are combined to improve their therapeutic effects, this is referred to as drug synergism [49]. For various diseases, including MM, appropriate drug combinations can reduce drug resistance or maximize efficacy [50,51,52,53]. For instance, the previously mentioned VRD is currently an effective and well-tolerated pre-ASCT induction protocol, benefiting newly diagnosed MM patients [6]. BTZ, as a clinically used protease inhibitor, is a cornerstone of VRD. However, significant obstacles to using BTZ to treat MM are drug resistance and the malignancy of MM. To cross these obstacles, researchers have investigated the BTZ resistance-reducing effects of various drug combinations and their synergistic anti-MM effects, such as daratumumab (anti-CD38 antibody), BTZ, and dexamethasone (CASTOR trial), TAK-243 (novel and specific UAE inhibitor) and BTZ, decitabine (epigenetic modulator) and BTZ [54,55,56]. Additionally, recent studies have demonstrated that some naturally occurring polyphenolic compounds have anti-MM activity. These polyphenolic compounds also have the potential to reduce BTZ resistance. Given the low toxicity of natural polyphenolic compounds, their synergistic combination with BTZ may offer MM patients extra therapeutic options [19,20].

## 6. Flavonoids and Non-Flavonoid Polyphenols in MM Therapy

Polyphenols are a class of phytochemicals divided into flavonoids and non-flavonoids. A group of compounds known as flavonoids consists of two benzene rings connected by phenolic hydroxyl groups by a central three-carbon atom. They are secondary metabolites in fruits, vegetables, and other herbal plants. The main flavonoid subgroup includes flavones, flavonols, flavan-3-ols, anthocyanins, isoflavones, and chalcones (Figure 4) [57]. The main non-flavonoids contained phenolic, hydroxycinnamic, lignans, stilbenes, and tannins [58]. Based on the results of this systematic review, various polyphenols, including flavonols(icariin, icariside II), flavan-3-ols((-)-epigallocatechin-3-gallate), flavone(scutellarein, wogonin, morin), isoflavone(formononetin, daidzin), plant extracts rich in flavonoids(punica granatum juice) and non-flavonoid polyphenols(silibinin, resveratrol, curcumin, caffeic acid) exerted anti-MM synergistic effects in combination with BTZ in vivo and/or in vitro(Table 2) (Figure 5).

Flavonols are a class of flavonoids with a 3-hydroxyflavonoid backbone. Two flavonols are of interest in synergistic anti-MM therapy. Icariin, an active ingredient in the stem and leaves of Epimedium, exerts significant anti-tumor effects on a variety of human tumor cells [22]. Similarly, icariside II, isolated from E. koreanum, inhibits the proliferation and induces apoptosis of many human cancer cells, such as MM, breast cancer, and prostate cancer [23].

Flavan-3-ols are flavan derivatives with a 2-phenyl-3,4-dihydro-2H-chroman-3-ol backbone. One flavan-3-ol is of interest in synergistic anti-MM therapy. (-)-epigallocatechin-3-gallate(EGCG) belongs to the category of catechins, commonly found in green tea. Catechins have anti-mutagenic, tumor progression-inhibiting, apoptosis-promoting, and antioxidant effects in various solid tumors [59,62].

Flavones are a class of flavonoids with a 2-phenylchromen-4-one backbone. Three flavones are of interest in synergistic anti-MM therapy. Scutellarein, obtained from *S. baicalensis* or *S. barbata*, exerts anticancer activity against various tumors, including breast, lymphoma, colorectal, lung, and liver cancers [24,25]. Similarly, wogonin, an active monoflavone from Scutellaria baicalensis with anti-angiogenic activity, is a potential anti-cancer drug with low toxicity [26]. Morin, isolated from members of the mulberry family, such as mulberry figs and old figs, has anti-proliferative activity against some tumors, such as oral squamous cell carcinoma, leukemia, and colorectal cancer [27].

In contrast to flavones, isoflavones have a chemical structure based on a 3-phenylchromium-4-one backbone. Two isoflavones are of interest in synergistic anti-MM therapy. Formononetin, mainly isolated from the roots of Astragalus membranaceus, Trifolium pratense, Glycyrrhiza glabra, and Pueraria lobate, has anti-inflammatory, antioxidant, antiviral, neuroprotective, wound healing, and antitumor biological activities [28,29]. Daidzin can be isolated from Pueraria lobate. It has demonstrated anticancer activity in preventing and treating breast and prostate cancers [30].

In addition, Punica granatum juice(PGJ), one plant extract rich in flavonoids, is also of attention. The drug activity of PGJ is related to many flavonoid phytoactive components, including the anthocyanins catechin, quercetin, kaempferol, apigenin, and lignan. The anticancer effects of PGJ are widely used to prevent and treat colon, lung, skin, and prostate cancers [31].

Finally, polyphenols other than flavonoids belong to non-flavonoid polyphenols. Four non-flavonoid polyphenols are of interest in synergistic anti-MM therapy. Silibinin, an extract of Milk Thistle, exerts very high antioxidant and antitumor properties [32,60]. Resveratrol is widely found in grapes, berries, and peanuts and has anticancer activity against most human cancers [61]. Another non-flavonoid polyphenol, curcumin, is present in turmeric root. It exerts anti-inflammatory, anti-atherosclerotic, and anti-tumor pharmacological effects [33]. In addition, caffeic acid, an active component of honeybee propolis, has cytotoxic, apoptosis, and anti-proliferation effects [34].

## 7. Synergistic Effects of Flavonoids and Bortezomib in Anti-MM

Nine flavonoids (icariin, icariside II, EGCG, scutellarein, wogonin, morin, formononetin, daidzin) plus a flavonoid-rich plant extract (*PGJ*) synergize with BTZ in anti-MM by regulating proliferation, apoptosis, and drug resistance-related signaling pathways (Figure 6) (Table 3).

### 7.1. Icariin and Bortezomib

In KM3/BTZ-resistant cells, icariin increased the sensitivity of KM3/BTZ cells to BTZ and partially reversed the drug resistance. Its effect on changing drug resistance may be mediated by upregulating the expression of pro-apoptotic cytokine Par-4, decreasing the expression of drug resistance proteins HSP27 and P-gp [22]. Another study found that the combination of icariin and BTZ promoted apoptosis in U266 cells by blocking the JAK/STAT pathway and lowering the expression of anti-apoptotic proteins like Bcl-2, Bcl-xl, and Survivin. This combination also delayed the cell cycle progression, as shown by the result that more U266 cells were in the G0/G1 phase [23].

### 7.2. Icariside II and Bortezomib

While icariside II enhanced the apoptotic effect of BTZ in U266 cells, it also inhibited the JAK/STAT pathway and down-regulated the expression of STAT3 target genes Bcl-2, Bcl-xl, Survivin, cyclin D1, COX-2, and VEGF, beneficially inhibiting proliferation and promoting apoptosis [35].

### 7.3. EGCG and Bortezomib

EGCG, a catechin in green tea, and BTZ synergistically inhibited KM3 cell growth and induced apoptosis by inhibiting NF-κB/P65 expression, down-regulating pIκBα, and up-regulating IκBα expression [36].

### 7.4. Scutellarein and Bortezomib

In vitro, scutellarein alone time-dependently reduced cell viability and significantly induced apoptosis in MM.1R and IM-9 cells. In vivo, dual intervention with scutellarein and BTZ significantly reduced xenograft tumor burden in nude mice with no significant effect on mouse body weight. In parallel, protein expression levels of some apoptotic markers were altered, such as active caspase-3 (upregulated), Bax (upregulated), and Bcl-2 (downregulated) [25]. In another in vivo study, scutellarein eliminated MM cell resistance to BTZ through multiple mechanistic pathways. This result was caused by the HDAC/miR-34a-mediated epigenetic regulation of the c-Met/Akt/mTOR pathway and the NF-κB-mediated activation of the apoptosis cascade [24].

### 7.5. Wogonin and Bortezomib

Wogonin and BTZ synergistically inhibited the secretion levels of pro-angiogenic factors VEGF, PDGF, and bFGF in RPMI 8226 cells. The angiogenesis-inhibitory effect of wogonin may be related to its inhibition of the c-Myc/HIF-1α signaling axis [26].

### 7.6. Morin and Bortezomib

Morin potentiates BTZ-induced apoptosis in U266 cells, from 18.7% to 51.2%, by inhibiting the STAT3 pathway, leading to downregulation of STAT3-dependent gene expressions, such as XIAP, cFLIP, Mcl-1, Survivin, Bcl2, BclXl, and c-IAP-2. And morin’s inhibition of STAT3 phosphorylation was more pronounced than other flavonols (galangin, kaempferol, quercetin, and myricetin) with different positions and numbers of its hydroxyl groups on the B-loop [27].

### 7.7. Formononetin and Bortezomib

In vitro, formononetin and BTZ enhanced STAT3 inhibition and promoted the death of U266 cells [28]. Subsequently, a follow-up study by the same research group found that formononetin and BTZ exert synergistic enhancement of anti-proliferation and pro-apoptosis by blocking the activation of NF-κB, PI3K/AKT, and AP-1 [29].

### 7.8. Daidzin and Bortezomib

Daidzin synergistically increased the apoptotic and cytotoxic effects of BTZ by inhibiting the activation of STAT3 and its upstream kinases (JAK1, JAK2, and c-Src). In addition, this drug combination increased caspase-3 activation and PARP cleavage, leading to the downregulation of the expression of various oncogenic apoptotic proteins [30].

### 7.9. Plant Extracts and Bortezomib

In vitro, PGJ inhibited angiogenesis, microvascular growth outside of aortic rings, cell migration, and invasion in MM cells. In addition, After BTZ exposure, PGJ intervention increased the cytotoxic effects on U266 cells [31].

## 8. Synergistic Effects of Non-Flavonoid Polyphenols and Bortezomib in Anti-MM

Four flavonoids (silibinin, resveratrol, curcumin, caffeic acid) synergize with BTZ in anti-MM effect by regulating proliferation, apoptosis, and drug resistance-related signaling pathways (Table 4).

### 8.1. Silibinin and Bortezomib

In combination with low concentrations of BTZ, silibinin increased the cytotoxic effect of BTZ by increasing the expression of activated caspases, which promoted apoptosis [32].

### 8.2. Resveratrol and Bortezomib

In vitro, resveratrol induced apoptosis in MM144 cells by upregulating the Fas/CD95 signaling pathway and caspase-8 and caspase-10. Moreover, when used in combination, resveratrol, and BTZ highly enhanced U266 cell apoptosis [37].

### 8.3. Curcumin and Bortezomib

Curcumin enhances the apoptotic effect of BTZ on MM cells by regulating multiple signaling pathways (Figure 7). The bone marrow microenvironment, in which bone marrow stromal cells (BMSCs) interact with MM, influences the survival and growth of MM cells. In vitro, curcumin inhibited the activation of JAK/STAT and MAPK pathways in U266 cells after treatment with BMSCs cell supernatant. Additionally, curcumin and BTZ co-treatment efficiently prevented IL-6-induced STAT3 and Erk phosphorylation, increased PARP cleavage, and decreased pro-caspase-3 levels. Through these mechanisms mentioned above, this combination inhibited the growth of U266 cells and promoted apoptosis [38]. After that, in another trial, curcumin enhanced the inhibitory effect of BTZ on NF-κB activation in U266 cells, leading to increased apoptosis. In vivo, the combination group was more potent than the BTZ group in reducing tumor volume [39]. Moreover, curcumin increased the expression of cleaved caspase-3 protein by inhibiting the Notch1 signaling pathway, which increased the drug sensitivity of RPMI-8226 cells and U266 cells to BTZ [33].

A class of novel curcumin analogs known as amino acid adducts of curcumin improved the proteasomal inhibitory effect of BTZ on MM cells and they enhanced the proliferation inhibition and apoptosis induction of BTZ. Notably, similar to curcumin, the twelfth curcumin analog increased PARP and caspase-3 cleavage, enhancing BTZ-induced apoptosis. The water-soluble, highly bioavailable twelfth curcumin analog has the potential to replace curcumin in anti-MM therapy in the future [40].

### 8.4. Caffeic acid and Bortezomib

In vitro, caffeic acid alone inhibits NF-κB-binding activity and IL-6 levels, which are closely linked to apoptosis and growth of tumor cells. Moreover, the combination of caffeic acid and BTZ synergistically increased cytotoxic and anti-proliferative effects on ARH-77 cells [34].

## 9. Conclusions and Future Directions

Although treatment options for MM continue to be optimized, the prognosis remains unsatisfactory. The aggressiveness and drug resistance of malignant tumors hinder the current treatment with protease inhibitors—especially BTZ. In this regard, flavonoids and non-flavonoid polyphenols are potential supportive therapies to address some of the challenges faced in BTZ treatment. Flavonoids(icariin, icariside II, EGCG, scutellarein, wogonin, morin, formononetin, daidzin), plant extract rich in flavonoids(PGJ), and non-flavonoid polyphenols(silibinin, resveratrol, curcumin, caffeic acid) combined with BTZ demonstrate the synergistic anti-MM effect. These synergistic anti-MM effects were achieved by anti-proliferative, pro-apoptotic, and anti-drug resistance. Based on those pieces of evidence, flavonoids, plant extract rich in flavonoids, and non-flavonoid polyphenols may benefit patients with MM, especially by overcoming the challenges faced in BTZ therapy.

According to some other relevant preclinical evidence, some polyphenols alone have anti-MM activity, such as isoginkgetin (a biflavone) and virola oleifera (a polyphenol-rich plant extract) [63,64]. Their synergistic effects, in combination with BTZ for the treatment of MM, need to be further explored. In addition, most of the synergistic anti-MM impacts of polyphenols + BTZ are only verified at the cellular level, such as icariin, icariside, EGCG, wogonin, morin, daidzin, PGJ, silibinin, and resveratrol. Their anti-MM synergistic effects in combination with BTZ need further clinically relevant animal models for validation. These polyphenols + BTZ exert synergistic effects through modulation of MM-related signaling pathways such as NF-κB, PI3K/Akt, Ras/Raf/MEK/Erk, JAK/STAT, and Wnt/β-catenin. However, the signaling pathways of each combination of anti-MM are still rudimentary, and more research is needed to refine the signaling pathway network. Although these results show promise for a synergistic MM treatment using these polyphenolic compounds and BTZ, the evidence at this time is only preclinical. Therefore, further clinical studies must demonstrate these synergistic effects. Moreover, preclinical studies on the anti-MM synergistic effects of these polyphenols with two other clinically available protease inhibitors, carfilzomib, and estazomib, need to be conducted to provide more options for clinical combination use. Adverse effects of protease inhibitor use are another challenge in the treatment of MM, although not within the scope of the review. Some polyphenols have the ameliorative potential for the non-tumor toxicity of protease inhibitors. For instance, rutin mitigates carfilzomib-induced cardiotoxicity by inhibiting NF-κB, mast gene expression, and reducing oxidative stress [65]. Resveratrol, by activating SIRT1, improves BTZ mechanical nociceptive sensitization [66]. For the comprehensive management of MM, additional studies on the antagonism of polyphenols against protease inhibitor-associated toxicity are also of clinical value. Finally, BTZ could alleviate MM-associated bone disease by regulating the RANK/RANKL/OPG signaling pathway [67]. By concentrating on the modulation of the RANK/RANKL/OPG signaling pathway, it is possible to investigate whether combining polyphenols and BTZ has beneficial effects on MM-associated bone disease.

Some polyphenols contain catechol moieties, such as EGCG, quercetin, and myricetin. Notably, they reduce the protease inhibitory activity of BTZ by forming stable cyclic boronic esters, thereby decreasing the anti-MM effect of BTZ [68,69]. A representative potential BTZ antagonistic polyphenol, EGCG, demonstrated dose-dependent inhibition of BTZ antitumor activity [62]. An in vitro study found that, by activating Wnt/β-catenin, EGCG antagonized the antitumor effect of BTZ [70]. Therefore, clinical staff members should be cautious about BTZ drug interactions with these polyphenols in clinical use. Last but not least, the problem of bioavailability is an important reason why polyphenols do not work as well as they could in the body. In the future, Low bioavailability polyphenols will eventually need to be combined with new drug delivery systems or other derivatives to utilize their anticancer capabilities fully.

## Figures and Tables

**Figure 1 biomolecules-12-01647-f001:**
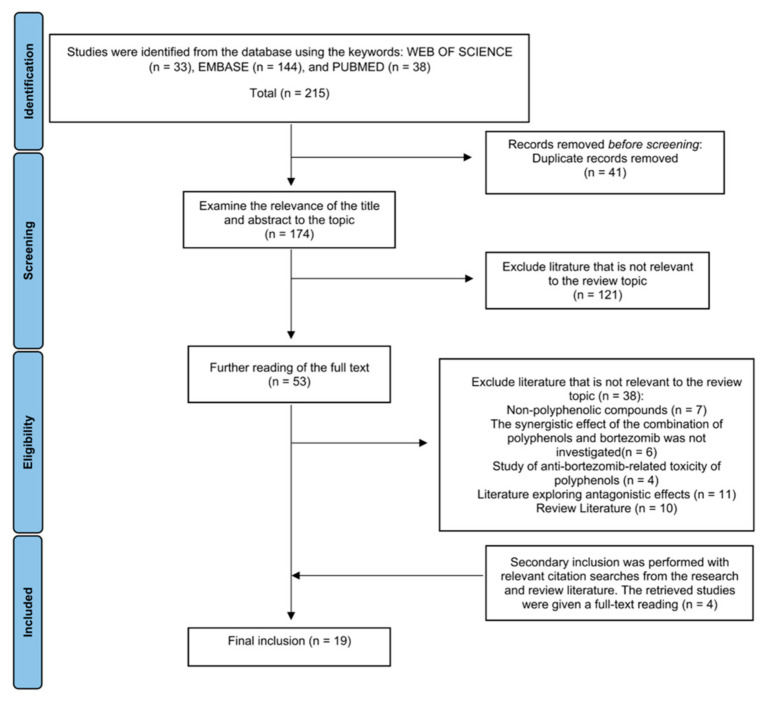
Diagram of the process for including and excluding the literature for this review.

**Figure 2 biomolecules-12-01647-f002:**
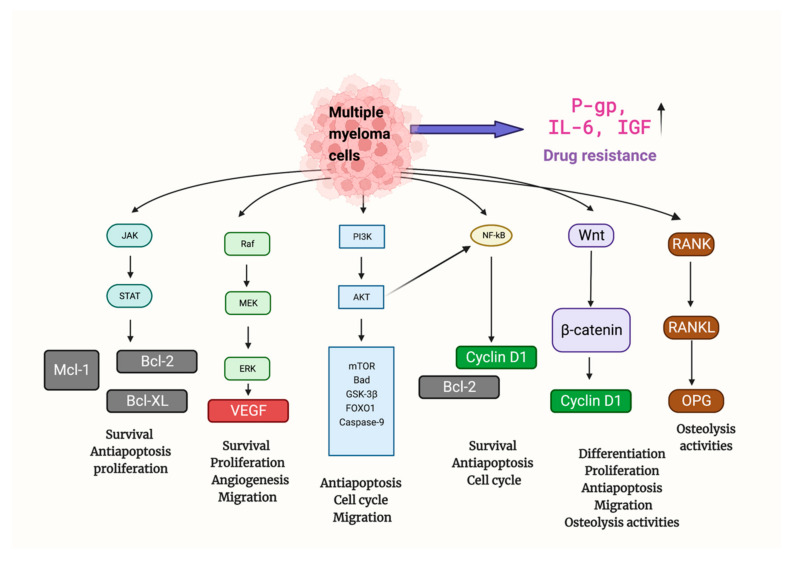
Multiple signaling pathways are involved in the tumor formation and progression of MM. Mechanisms promoting anti-apoptosis, proliferation, angiogenesis, migration, and drug resistance were upregulated, while mechanisms related to apoptosis and normal cell cycle regulation were suppressed. Abbreviations: Bad: BCL2 associated agonist of cell death; Bcl2: B-cell lymphoma 2; Bcl-xl: B-cell lymphoma-extra-large; FOXO1: Forkhead box protein O1; GSK-3β: Glycogen synthase kinase-3 beta; JAK/STAT: Janus kinases/signal transducer and activator of transcription proteins; Mcl-1: Myeloid cell leukemia 1; MEK/Erk: MAP kinase–ERK kinase/extracellular-signal-regulated kinases; TOR: mammalian target of rapamycin; PI3K/Akt: Phosphoinositide 3-kinases/Protein Kinase B; RANK/RANKL/OPG: Receptor activator of nuclear factor κB/Receptor activator of nuclear factor kappa-Β ligand/Osteoprotegerin; Wnt: Wingless and Int-1.

**Figure 3 biomolecules-12-01647-f003:**
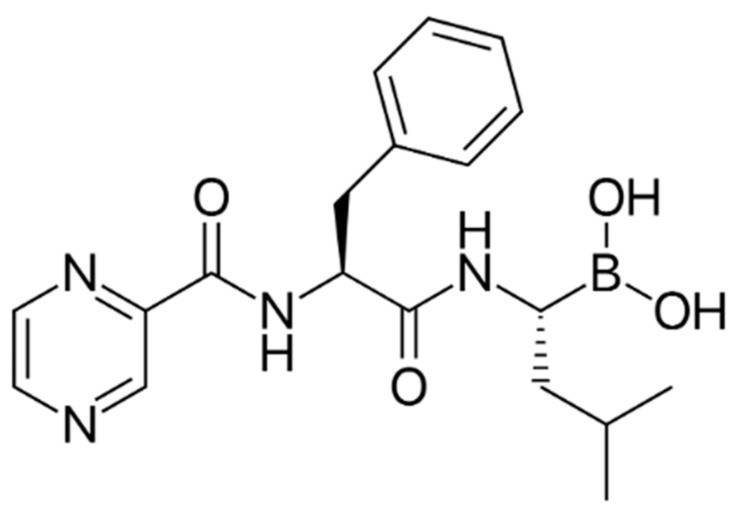
Chemical structure of bortezomib, an anti-MM Protease inhibitor.

**Figure 4 biomolecules-12-01647-f004:**
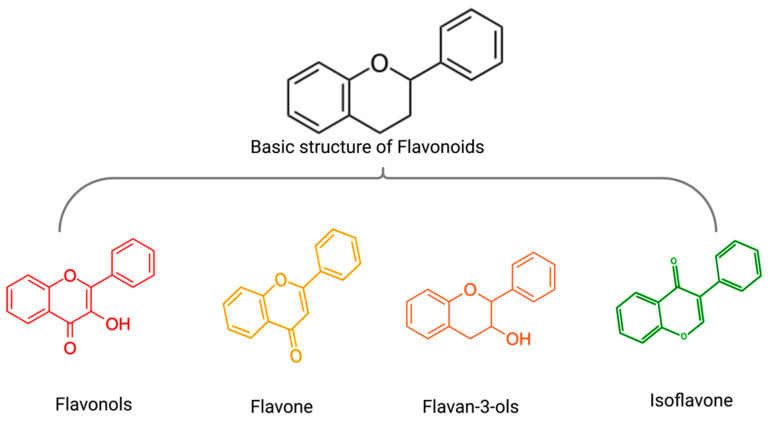
The basic structure of flavonoids(black) and the general structure of flavonols (red), flavone (yellow), flavan-3-ols (orange), isoflavone (green).

**Figure 5 biomolecules-12-01647-f005:**
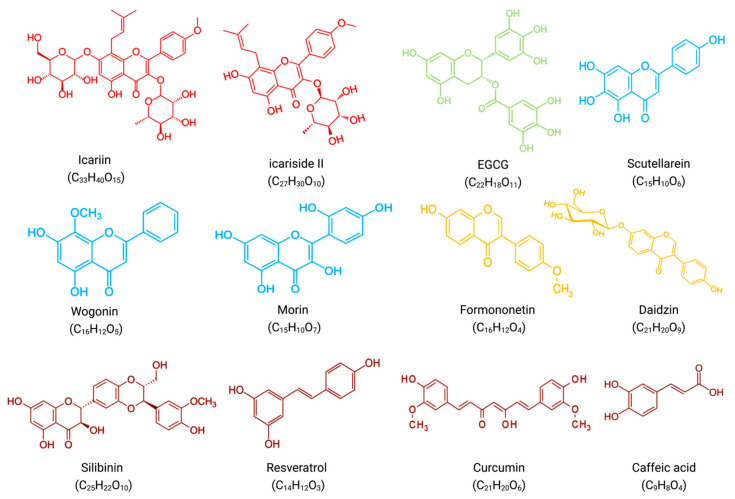
Molecular formulae and chemical structures of flavonoid compounds and non-flavonoid polyphenolic compounds(excluding PGJ, a plant extract rich in flavonoids). Flavonols(icariin, icariside II) have a red chemical formula. Flavan-3-ol (EGCG) is green in color. The formula of flavones (scutellarein, wogonin, morin) is blue. The chemical formula of isoflavones (formononetin, daidzin) is yellow. Non-flavonoid polyphenols (silibinin, resveratrol, curcumin, caffeic acid) have a brown chemical formula.

**Figure 6 biomolecules-12-01647-f006:**
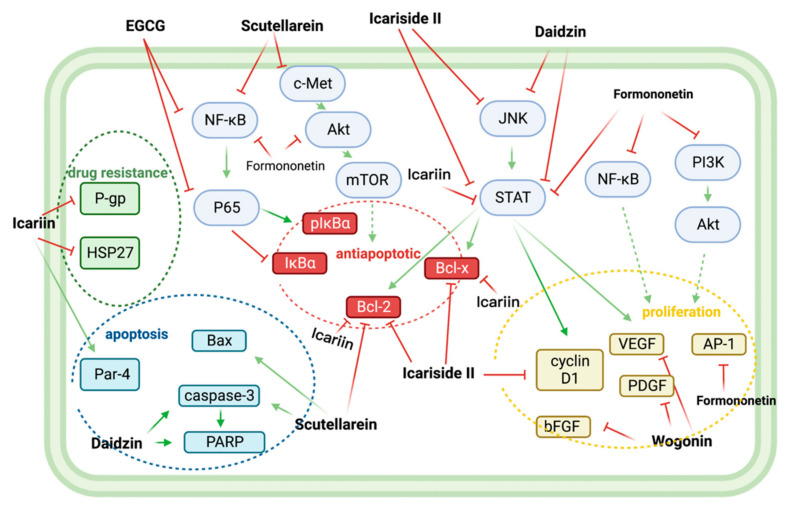
Nine flavonoids, icariin, icariside II, EGCG, scutellarein, wogonin, morin, formononetin, daidzin, and isoginkgetin, together exerted multiple synergistic BTZ anti-MM effects. Icariin synergistically improved the drug sensitivity of MM to BTZ by reducing the expression of drug resistance proteins HSP27 and P-gp. EGCG, scutellarein, icariin, icariside II, and formononetin down-regulated the downstream anti-apoptotic proteins Bcl2, BclXl, pIκBα by inhibiting NF-κB, c-Met/Akt/mTOR, and JAK/STAT pathways, thereby synergistically reversing anti-apoptosis. Icariside II, formononetin, and Wogonin, with BTZ synergistically, exerted antiproliferative effects by downregulating VEGF, PDGF, bFGF, cyclin D1, and AP-1. Finally, scutellarein, Daidzin, and Icariin enhanced BTZ-related apoptosis by upregulating the expression of Par-4, Bax, caspase-3, and PARP. Abbreviations: AP-1: Activator protein 1; Bax: bcl-2-like protein 4; Bcl2: B-cell lymphoma 2; Bcl-xl: B-cell lymphoma-extra-large; bFGF: basic fibroblast growth factor; c-Met: tyrosine-protein kinase Met; NF-κB/P56: Nuclear factor-kappa-B/p65; HSP27: Heat shock protein 27; JAK/STAT: Janus kinases/signal transducer and activator of transcription proteins; IκBα: nuclear factor of kappa light polypeptide gene enhancer in B-cells inhibitor, alpha; TOR: mammalian target of rapamycin; PARP: Poly (ADP-ribose) polymerase; Par-4: Prostate apoptosis response-4; PDGF: Platelet-derived growth factor; P-gp: P-glycoprotein; PI3K/Akt: Phosphoinositide 3-kinases/Protein Kinase B.

**Figure 7 biomolecules-12-01647-f007:**
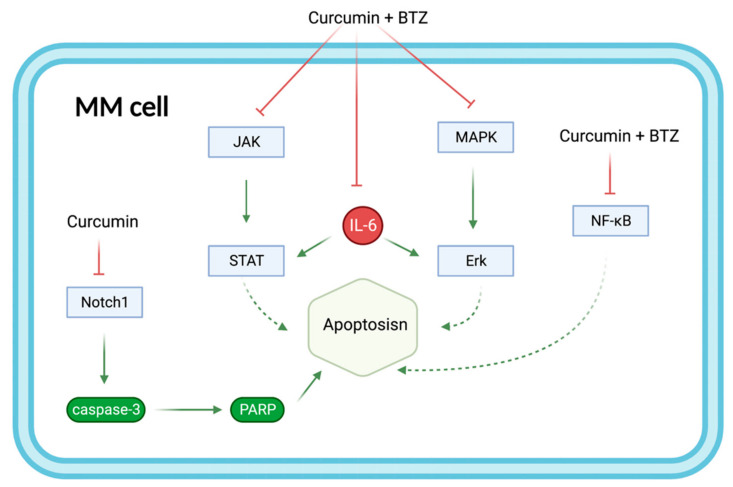
The combination of curcumin and BTZ synergistically enhances the effect on MM apoptosis by regulating multiple signaling pathways. Curcumin+BTZ synergistically enhanced apoptosis through potent inhibition of IL-6-induced JNK/STAT and MAPK/ERK, as well as NF-κB. Additional inhibition of the Notch1 signaling pathway by curcumin increased the expression of caspase3 protein and PARP and therefore enhanced apoptosis. Abbreviations: BTZ: bortezomib; NF-κB: Nuclear factor-kappa-B; MAPK/ERK: mitogen-activated protein kinases/extracellular signal-regulated kinases; MM: Multiple myeloma; Notch1: Neurogenic locus notch homolog protein 1; JAK/STAT: Janus kinases/signal transducer and activator of transcription proteins; PARP: Poly (ADP-ribose) polymerase.

**Table 1 biomolecules-12-01647-t001:** General information on the 19 publications included in this systematic review.

Authors	Year	Title	Polyphenols, or Plant Extracts Rich in Polyphenols	Journal Title, Volume and Page Number	Doi
Li, Z. Y., et al. [22]	2017	Icaritin Reverses Multidrug Resistance of Multiple Myeloma Cell Line KM3/BTZ.	Icariin	Zhongguo shi yan xue ye xue za zhi 25(6): 1690–1695.	10.19746/J.CNKI.ISSN.1009-2137.2019.02.025
Kim, S. H., et al. [23]	2011	Janus activated kinase 2/signal transducer and activator of transcription 3 pathway mediates icariside II-induced apoptosis in U266 multiple myeloma cells.	Icariside II	Eur J Pharmacol 654(1): 10–16.	10.1016/J.EJPHAR.2010.11.032
Li, L., et al. [24]	2020	Scutellarin circumvents chemoresistance, promotes apoptosis, and represses tumor growth by HDAC/miR-34a-mediated down-modulation of Akt/mTOR and NF-κB-orchestrated signaling pathways in multiple myeloma.	Scutellarein	Int J Clin Exp Pathol 13(2): 212–219.	Not available
Shi, L., et al. [25]	2019	Scutellarein selectively targets multiple myeloma cells by increasing mitochondrial superoxide production and activating intrinsic apoptosis pathway.	Scutellarein	Biomed Pharmacother 109: 2109–2118.	10.1016/j.biopha.2018.09.024
Fu, R., et al. [26]	2016	Wogonin inhibits multiple myeloma-stimulated angiogenesis via c-Myc/VHL/HIF-1α signaling axis.	Wogonin	Oncotarget 7(5): 5715–5727.	10.18632/ONCOTARGET.6796
Gupta, S. C., et al. [27]	2013	Morin inhibits STAT3 tyrosine 705 phosphorylation in tumor cells through activation of protein tyrosine phosphatase SHP1.	Morin	Biochem Pharmacol 85(7): 898–912.	10.1016/J.BCP.2012.12.018
Kim, C., et al. [28]	2018	Formononetin-induced oxidative stress abrogates the activation of STAT3/5 signaling axis and suppresses the tumor growth in multiple myeloma preclinical model.	Formononetin	Cancer Lett 431: 123–141.	10.1016/j.canlet.2018.05.038
Kim, C., et al. [29]	2019	Formononetin Regulates Multiple Oncogenic Signaling Cascades and Enhances Sensitivity to Bortezomib in a Multiple Myeloma Mouse Model.	Formononetin	Biomolecules 9(7): 262.	10.3390/biom9070262
Yang, M. H., et al. [30]	2019	Attenuation of STAT3 Signaling Cascade by Daidzin Can Enhance the Apoptotic Potential of Bortezomib against Multiple Myeloma.	Daidzin	Biomolecules 10(1): 23	10.3390/biom10010023
Tibullo, D., et al. [31]	2016	Antiproliferative and Antiangiogenic Effects of Punica granatum Juice (PGJ) in Multiple Myeloma (MM).	PGJ	Nutrients 8(10):611.	10.3390/nu8100611
Geraldes, C., et al. [32]	2015	New targeted drugs in multiple myeloma therapy-in vitro studies.	Silibinin	Clinical Lymphoma, and Leukemia 15: e254.	10.1016/j.clml.2015.07.539
Ge, X. P., et al. [33]	2019	Curcumin Increases the Chemosensitivity of Multiple Myeloma to Bortezomib by Inhibiting the Notch1 Signaling Pathway.	Curcumin	Zhongguo shi yan xue ye xue za zhi 27(2): 464–471.	10.19746/j.cnki.issn.1009-2137.2019.02.025
Altayli, E., et al. [34]	2015	An in vitro and in vivo investigation of the cytotoxic effects of caffeic acid (3,4-dihydroxycinnamic acid) phenethyl ester and bortezomib in multiple myeloma cells.	Caffeic acid	Turk J Med Sci 45(1): 38–46.	10.3906/SAG-1401-127
Jung, Y. Y., et al. [35]	2018	Anti-myeloma Effects of Icariin Are Mediated Through the Attenuation of JAK/STAT3-Dependent Signaling Cascade.	Icariin	Front Pharmacol 9: 531.	10.3389/fphar.2018.00531
Wang, Q., et al. [36]	2009	Potentiation of (-)-epigallocatechin-3-gallate-induced apoptosis by bortezomib in multiple myeloma cells.	EGCG	Acta Biochim Biophys Sin (Shanghai) 41(12): 1018–1026.	10.1093/ABBS/GMP094
Reis-Sobreiro, M., et al. [37]	2009	Involvement of mitochondria and recruitment of Fas/CD95 signaling in lipid rafts in resveratrol-mediated antimyeloma and antileukemia actions.	Resveratrol	ONCOGENE 28(36): 3221–3234.	10.1038/onc.2009.183
Park, J., et al. [38]	2008	Curcumin in combination with bortezomib synergistically induced apoptosis in human multiple myeloma U266 cells.	Curcumin	Mol Oncol 2(4): 317–326.	10.1016/J.MOLONC.2008.09.006
Sung, B., et al. [39]	2009	Curcumin circumvents chemoresistance in vitro and potentiates the effect of thalidomide and bortezomib against human multiple myeloma in nude mice model.	Curcumin	Mol Cancer Ther 8(4): 959–970.	10.1158/1535-7163.MCT-08-0905
Mujtaba, T., et al. [40]	2012	Sensitizing human multiple myeloma cells to the proteasome inhibitor bortezomib by novel curcumin analogs.	Novel curcumin analogs	Int J Mol Med 29(1): 102–106.	10.3892/ijmm.2011.814

**Table 2 biomolecules-12-01647-t002:** Classes and sources of 13 polyphenols (9 flavonoids and 4 non-flavonoids) that synergize with BTZ against MM.

Polyphenol	Class	Source	Reference
Icariin	Flavonol	Dried stems and leaves of arrow leaf Epimedium, pilose Epimedium, Wushan Epimedium, Korean Epimedium.	[22]
icariside II	Flavonol	Stems and leaves of Epimedium koreanum Nakai (Berberidaceae).	[23]
(-)-epigallocatechin-3-gallate	Flavan-3-ol	Green tea	[59]
Scutellarein	Flavone	*S. baicalensis* or *S. barbata*	[24,25]
Wogonin	Flavone	Scutellaria baicalensis	[26]
Morin	Flavone	Mulberry figs and old fustic (Chlorophora tinctoria)	[27]
Formononetin	Isoflavone	Roots of Astragalus membranaceus, Trifolium pratense, Glycyrrhiza glabra, and Pueraria lobate	[28,29]
Daidzin	Isoflavone	Soy and soy products	[30]
Punica granatum Juice	Plant extract rich in polyphenols	Punica granatum	[31]
Silibinin	Non-flavonoid polyphenol	Silybum marianum	[32,60]
Resveratrol	Non-flavonoid polyphenol	Grape, blueberry, raspberry, and mulberries peel and peanut	[61]
Curcumin	Non-flavonoid polyphenol	Turmeric rhizome	[33]
Caffeic acid	Non-flavonoid polyphenol	Honey Bee Propolis	[34]

**Table 3 biomolecules-12-01647-t003:** Mechanistic anti-MM effects of flavonoid-BTZ combinations, as demonstrated in vitro and in vivo.

Flavonoid	Cell Line	Flavonoid Concentration or Dose	Bortezomib Concentration or Dose	Effects	Reference
Icariin	Drug-resistant cell line KM3/BTZ	0, 5, 10, 20, 40 mol/L	0.08, 0.16, 0. 32, 0.64, 1.28 μg/mL	Increased drug sensitivity and reversed drug resistance	[22]
Icariin	U266	10 μM	1 nM	Enhanced cytotoxic effect, G0/G1 phase cell cycle arrest, and apoptosis	[35]
Icariside II	U266	0, 25, 50, 100 μM	20, 40 nM	Enhance cell apoptosis	[23]
EGCG	KM3 cell line	0, 25, 50, 100 µM	20 nM	Synergistically inhibit cell growth and induce cell apoptosis	[36]
Scutellarein	MM.1R, IM-9, U266B1, RPMI 8226 IM-9 cells injected subcutaneously in BALB/C nude mice	200 μg/mL 0, 30, 60, 120 mg/kg	30, 60 mg/kg	Inhibition of cell viability and selective induction of cell apoptosis. The tumor burden of xenograft in nude mice was significantly reduced, and body weight was not affected.	[25]
Scutellarein	MM.1S cells injected subcutaneously in the BALB/C nude mice	30, 60 mg/kg	30, 60 mg/kg	The tumor weight was significantly reduced and Pharmacochemical resistance was reduced	[24]
Wogonin	RPMI 8226, U266	20, 40, 80 μM	10 nM	Synergistically inhibit tumor-stimulated angiogenesis.	[26]
Morin	U266, RPMI 8226, MM.1S	5, 10, 25 µM	20 nmol/L	Enhanced drug-induced apoptosis	[27]
Formononetin	U266	50 μM	10 nM	Enhanced drug-induced apoptosis	[28]
Formononetin	U266, RPMI 8226 U266 cells injected subcutaneously in the athymic nu/nu female mice	50 μM 20 mg/kg	10 nM 0.25 mg/kg	Synergistically enhanced anti-tumor and apoptotic effects	[29]
Daidzin	U266, MM1.S	10, 20, 30 µM	1, 2.5, 5 nM	Decreased cell growth, induced higher cell death, increased SubG1 arrest and increased cell apoptosis.	[30]
Punica granatum juice	KMS26, MM1S, U266	3, 6, 12%	15 nM	Improved the cytotoxic effect	[31]

**Table 4 biomolecules-12-01647-t004:** Mechanistic anti-MM effects of non-flavonoid polyphenols-BTZ combinations, as demonstrated in vitro and in vivo.

Non-Flavonoid Polyphenol	Cell Line	Non-Flavonoid Polyphenol Concentration or Dose	Bortezomib Concentration or Dose	Effects	Reference
Silibinin	NCI H929	Not available	Not available	Increased cytotoxic effects	[32]
resveratrol	MM1S, MM144, U266	10 mM	10 nM	Enhanced cell apoptosis	[37]
Curcumin	U266	4, 8 μM	0.5, 4 nM	Synergistically inhibit cell growth and promote apoptosis	[38]
Curcumin	U266, RPMI-8226, RPMI-8226-Dox-6, RPMI-8226-LR-5, MM.1S, MM.1R U266 cells injected subcutaneously into mice	5,10 μmol/L 1 g/kg	0.25 mg/kg, 100 μL	Enhanced anti-tumor effect	[39]
Curcumin	RPMI-8266, U266	0, 20, 40 μM	5 nmol/L	Enhanced cell apoptosis	[33]
novel curcumin analogs	RPMI-1640	2.5, 5, 10 μM	10 nM	Enhanced cell proliferation inhibition, apoptosis induction and proteasome inhibition	[40]
caffeic acid	ARH-77	5, 10, 20, 40, 80, 160 g/mL	1, 10, 20, 30, 50, 100 nM	Synergistically enhance Cytotoxic effects and anti-proliferation	[34]

## Data Availability

Not applicable.

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
