# Peer review of "Synergistically Anti-Multiple Myeloma Effects: Flavonoid, Non-Flavonoid Polyphenols, and Bortezomib"

_biomolecules, 2022, doi:10.3390/biom12111647_

Round 1
Reviewer 1 Report
The review entitled "Synergistically anti-multiple myeloma effects: flavonoid, non-flavonoid polyphenols, and bortezomib" is quite informative. Yet, it is not flawless and could be further improved.
I suggest to explain the review methodology, focusing on search databases and engines, filters applied, etc. maybe using PRISMA flow diagram.
Why don't the authors consider the last 2 years in their literature search?
Several recent studies are published that need to be taken into account to get an up-to-date picture.
Minor points:
Several typos need to be fixed. Capital letters for compounds are used randomly.
Journal format is not followed (style of references etc ...)
Due to many acronyms, an abbreviation list is required.
Author Response
Response to Reviewer 1
Firstly, we'd like to extend our gratitude to reviewer 1 for his or her suggestions for improvements and comments on this manuscript. Below is our response to each of these comments.
Point 1: I suggest to explain the review methodology, focusing on search databases and engines, filters applied, etc. maybe using PRISMA flow diagram.
Response 1: We are grateful to Reviewer 1 for proposing Point 1. We have added details to the review methodology section, and we have drawn a PRISMA flow diagram consistent with it.
Point 2: Why don't the authors consider the last 2 years in their literature search? Several recent studies are published that need to be taken into account to get an up-to-date picture.
Response 2: We thank Reviewer 1 for suggesting that this paper should focus on relevant research in real-time. For this suggestion, we rechecked PubMed, Web of Science, and Embase for preclinical studies related to this review (limited to 2019-2022). Three recent papers were retrieved(paper 1: https://doi.org/10.21873/anticanres.15536, paper 2: doi: , paper 3: doi: 10.1128/MCB.00489-18).
In paper 1, Matsuda et al. investigated the antagonistic effects of caffeic acid and chlorogenic acid on bortezomib neurotoxicity. In paper 2, Lu et al. compared the growth inhibitory potency of Sencha Tea Extracts and Bortezomib on 9 MM cell lines. In paper 3, Tsalikis investigated the cytotoxicity of Isoginkgetin on six different MM cell lines (MM.1S, OPM2, 8826, H929, JJN3, and U226). However, experiments combining Isoginkgetin and Bortezomib were not performed.
Given those, these studies were relevant to the topics we reviewed, but did not meet the inclusion criteria for this review. Therefore, we do not discuss these studies in the central part of the article. Instead, we cite these studies in the future directions section, discussing their potential relevance to this topic and their shortcomings.
Point 3: Several typos need to be fixed. Capital letters for compounds are used randomly.
Response 3: Spelling mistakes have been corrected. And lowercase has been used in place of the inappropriate capitalization.
Point 4: Journal format is not followed (style of references etc ...)
Response 4: We made changes to the manuscript in accordance with the journal format, particularly with regard to the references' citation style. Along with this, reference citation formats of table 2, table 3, and table 4 have also been altered.
Point 5: Due to many acronyms, an abbreviation list is required.
Response 5: With regard to all the abbreviations used in the manuscript, we have produced an abbreviation table.
Additional information: general information about the 19 publications included in this systematic review is compiled in Table 1, hoping to facilitate readers' access to these publications.
We have re-uploaded the revised and newly added figures and tables.
Reviewer 2 Report
Reviewer’s Comments to the Authors
In the review by Ding et al., entitled “Synergistically anti-multiple myeloma effects: flavonoid, non-flavonoid polyphenols, and bortezomib” authors described concisely and very aptly the synergistic role of various phytochemicals with bortezomib against multiple myeloma. This is a timely review and compile interesting findings. However, there remains certain concerns to be addressed before it can be considered for publication.
1. Authors described methodology for the review, that indicates that this is a Systematic review rather than a descriptive review. However, authors didn’t mention the same anywhere, not the protocol (e.g. PRISMA) has been mentioned. This should be specified. Moreover, the selection criteria should be shown in a figure as per systematic review guidelines.
2. Appropriate use of abbreviations should eb followed. Full forms should be mentioned at the first instances.
3. Many grammatical and typo errors have been observed. The manuscript should be revised accordingly.
4. Structure of each of the polyphenols/flavonoids/non-flavonoids may be added in the tables in a new column inserted next to the name of the phytochemicals.
5. There doesn't seem to be a flow to the section layouts. For example, after introduction and methodology section: Mechanisms of MM > Bortezomib in MM therapy > Flavonoids and non-flavonoid polyphenols in MM therapy followed by the next sections would be more appropriate. Please revise accordingly.
6. A section on drug synergism and its importance in current disease scenario would be interesting.
Author Response
Response to Reviewer 2
We appreciate the time and effort that Reviewer 2 has devoted to this manuscript. Our response to each comment is as follows.
Point 1: Authors described methodology for the review, that indicates that this is a Systematic review rather than a descriptive review. However, authors didn’t mention the same anywhere, not the protocol (e.g. PRISMA) has been mentioned. This should be specified. Moreover, the selection criteria should be shown in a figure as per systematic review guidelines.
Response 1: We appreciate reviewer 2 for bringing up point 1. Indeed, our article is a systematic review. We have added the research methodology section in detail. Furthermore, according to the guidelines of the PRISMA statement, we have drawn Figure 1.
Point 2: Appropriate use of abbreviations should eb followed. Full forms should be mentioned at the first instances.
Response 2: We rechecked all abbreviations to ensure that they were defined when the full name of the corresponding term first appeared. Considering this manuscript contains lots of abbreviations, we've compiled all the abbreviations and definitions in the text in a table of abbreviations. And we hope that this abbreviations table will provide easier reading for readers.
Point 3: Many grammatical and typo errors have been observed. The manuscript should be revised accordingly.Structure of each of the polyphenols/flavonoids/non-flavonoids may be added in the tables in a new column inserted next to the name of the phytochemicals.
Response 3: We have examined the entire manuscript meticulously. And we have made the appropriate corrections for identifying grammatical and spelling errors.
Point 4: Structure of each of the polyphenols/flavonoids/non-flavonoids may be added in the tables in a new column inserted next to the name of the phytochemicals.There doesn't seem to be a flow to the section layouts. For example, after introduction and methodology section: Mechanisms of MM > Bortezomib in MM therapy > Flavonoids and non-flavonoid polyphenols in MM therapy followed by the next sections would be more appropriate. Please revise accordingly.
Response 4: All the chemical formulas and molecular structures of the polyphenolic compounds—both flavonoid and non-flavonoid—discussed in this systematic review are shown in figure 5 that we have created.
Point 5: There doesn't seem to be a flow to the section layouts. For example, after introduction and methodology section: Mechanisms of MM > Bortezomib in MM therapy > Flavonoids and non-flavonoid polyphenols in MM therapy followed by the next sections would be more appropriate. Please revise accordingly.
Point 6: A section on drug synergism and its importance in current disease scenario would be interesting.
Response 5 and 6: We especially appreciate your mentioning points 5 and 6. These two comments made us realize that the line flow of the manuscript is not so smooth. Based on your suggestion, we have added a new section, "Exploration of synergistic drug combinations in MM therapy," which describes the definition of drug synergism and explains the importance of the appropriate use of drug synergism in the treatment of multiple myeloma.
Subsequently, we adjusted the line flow of the manuscript. Eventually, after deep thoughts and discussions, the layout of the sections after the introduction and methodology section was determined as follows: Mechanisms of MM > Bortezomib in MM therapy > Exploration of synergistic drug combinations in MM therapy > Flavonoids and non-flavonoid polyphenols in MM therapy > Synergistic effects of flavonoids and bortezomib in anti-MM > Synergistic effects of non-flavonoid polyphenols and bortezomib in anti-MM > Conclusion and future directions.
Additional information: general information about the 19 publications included in this systematic review is compiled in Table 1, hoping to facilitate readers' access to these publications.
We have re-uploaded the revised and newly added figures and tables.
Round 2
Reviewer 1 Report
The authors improved their manuscript based on the reviewer's comments. All suggestions have been taken into consideration and all questions have been satisfactorily answered.
The paper can be accepted as it is.
Author Response
Point 1: The authors improved their manuscript based on the reviewer's comments. All suggestions have been taken into consideration and all questions have been satisfactorily answered.
Response 1: We again appreciate reviewer 1 for his or her time and effort in reviewing this manuscript. And we sincerely send our best wishes to you.
Reviewer 2 Report
The authors have addressed all the comments satisfactorily in the revised version of the manuscript.
However, the manuscript does not contain the newly added Table 1 in the file. It is recommended to add the Table in the manuscript for further consideration.
Author Response
The authors have addressed all the comments satisfactorily in the revised version of the manuscript.
However, the manuscript does not contain the newly added Table 1 in the file. It is recommended to add the Table in the manuscript for further consideration.
We have inserted table 1 in the revised manuscript. We are very grateful for your remarkable comments on our manuscript, which have helped us improve our manuscript's quality and scientific validity. Finally, we appreciate your time and effort in reviewing this manuscript and send our best wishes to you.
